# BENCHMARKING SURVIVAL MODELS: TREATMENT EFFECTS, BIAS, AND EQUITY

## ABSTRACT

Survival models are widely used to model *time-to-event* or *survival* data, which represents the duration until an event of interest occurs. In clinical research, survival analysis is used for estimating the effects of treatments on patient health outcomes. Recent advancements in machine learning (ML) have aimed to improve survival analysis methods, but current evaluation practices largely focus on predictive performance, often neglecting critical factors such as the ability to accurately estimate treatment effects and possible consequences on health equity. Estimating treatment effects from time-to-event data presents unique challenges due to the complex problem setting, the extensive assumptions required for causal inference, biased observational data, and the ethical consequences of using model outcomes in real-world health decisions. In this work, we introduce a comprehensive benchmarking framework designed to evaluate survival models on their ability to estimate treatment effects under realistic conditions and in the presence of potential inequalities. We formalize the discussion of bias in survival modelling, identifying key sources of inequity, and outline practical desiderata for methods that model time-to-event treatment effects. We clarify common assumptions in survival analysis, discuss critical shortcomings in current evaluation practices, and propose a new benchmarking metric that can be used to better evaluate model calibration. Using this framework, we systematically compare traditional and modern survival models across multiple synthetic and real world datasets, investigating, among other challenges, model performance under mis-specification and observational biases. Through this benchmark, we provide actionable insights for researchers to develop more robust and equitable survival models.

## 1 INTRODUCTION

With the rise of methods capable of processing large electronic health records (EHR) datasets, there is growing excitement about using machine learning (ML) to extract new insights from existing medical data. In particular, observational data could be used to assess the potential impact of various clinical treatments on health outcomes, given personalized patient health data Liu et al. (2021); Tan et al. (2021). The focus of such analyses typically centers on *time-to-event* data (or *survival* data), representing the duration of time until a patient experiences a relevant health outcome (Klein and Moeschberger, 1997; Tutz and Schmid, 2016; Hernan and Robins, 2023). For example, clinical trials investigating cancer treatment often evaluate efficacy based on survival duration or time until disease progression (Reck et al., 2016; Mok et al., 2019). Survival data may come from a randomized controlled trial (RCT) or observational datasets, such as EHRs, which create additional modelling complications (Hernan and Robins, 2023). While many ML for healthcare works focus on predicting survival times directly (Huang et al., 2023), our interest lies in methods for estimating treatment-specific survival models, such as survival or hazard functions, which can be used to determine *treatment effects* (i.e. a conclusion that a treatment definitively impacts patient outcomes)–quantities critical to clinical decision-making (Singh and Mukhopadhyay, 2011; Faraone, 2008).

Despite the increased adoption of ML for survival analysis and treatment effects estimation, little attention has been given to proper benchmarking and evaluation methods. Estimating treatment effects from time-to-event data poses unique challenges, due to inherent complications such as *censoring*, the potential for biases, as well as the causal assumptions required for identifiability(Hernan and

Robins, 2023). Many survival methods rely on restrictive modelling assumptions that may not hold in practice. As treatment effect estimates can influence real-world medical decisions, a thorough understanding of the methodology is crucial, especially of possible impacts on health equity. Key challenges include: (1) complex data generating scenarios, (2) modelling assumptions violations, and (3) sources of model bias (4) potential sources of inequity. In this paper, we propose a benchmarking framework for the evaluation of survival methods used to estimate heterogeneous treatment effects from time-to-event data, addressing these key challenges.

**Related work.** Survival models can be categorized into 1) classical statistical survival models, which may be parametric, such as the logistic hazard model (Tutz and Schmid, 2016), semi-parametric, such as the Cox proportional hazards model (CoxPH) (Cox, 1972), and non-parametric, such as the Kaplan-Meier method (Kaplan and Meier, 1958); and 2) modern ML survival models, including tree-based and neural network approaches (Wang et al., 2017). ML methods such as neural networks (Nagpal et al., 2021; 2020; Katzman et al., 2016), random forests (Ishwaran et al., 2008; Cui et al., 2020), and Gaussian processes (Fernandez et al., 2016; Alaa, 2017) have also been applied to survival analysis. The Cox PH is widely used to estimate hazard ratios in clinical and epidemiological research, but its causal interpretation has faced controversy due to common methodological flaws and assumption violations in practice (Hernán, 2010; Martinussen et al., 2020; Martinussen, 2021). Tutz and Schmid (2016) provides a detailed discussion of discrete-time survival methods, Wang et al. (2017) provides a review of machine learning for survival analysis, and Wiegrebe et al. (2023) reviews deep learning methods specifically. We provide details on notable approaches in Appendix A.1.

Existing benchmarks or comparison studies of ML (and classical) methods for survival analysis have focused on empirically evaluating models on predictive ability (Zhang et al., 2021; Spooner et al., 2020), rather than fidelity to the ground truth hazard or survival models, which are necessary to estimate treatment effects. These works also do not investigate model performance in the presence of assumption violations, known biases (such as confounding or informative censoring), or impacts on health equity. Works proposing new methods for survival analysis often investigate model performance over few synthetic scenarios, which cannot comprehensively inform model behavior (Katzman et al., 2016; Cui et al., 2020). Related benchmarking works in the realm of ML for health have focused on estimation of continuous treatment effects (Curth et al., 2021b; Crabbe et al., 2022) or fairness in medical imaging (Zong et al., 2022). To our knowledge, our paper is the first to systematically evaluate survival models from a causal perspective.

**Contributions.** 1) We provide a comprehensive discussion and formalization of key biases and challenges that arise in survival analysis, particularly in the context of treatment effect estimation. These include biases due to confounding, informative censoring, and model mis-specification, with a focus on impacts on health equity. 2) We introduce a benchmarking framework, including a novel evaluation metric, designed to evaluate the ability of survival models to estimate heterogeneous, time-varying treatment effects in the face of various complications. 3) Through extensive experiments on both synthetic and real-world datasets, we provide critical insights into the performance of traditional and modern survival models as well as a guide for improved benchmarking practices.

## 2 TREATMENT EFFECTS FOR TIME-TO-EVENT DATA

**Problem setting.** *Time-to-event data*, or *survival data*, represents the duration until an event occurs. We assume access to a dataset $\mathcal{D} = \{x_i, a_i, y_i, \delta_i\}_{i=1}^N$, with $N$ drawn from baseline distribution, $\mathbb{P}_0$. $X \in \mathbb{R}^d$ represents patient covariates, and $A \sim \{0, 1\}$ is the assigned treatment. *Right censoring*, when patient data is unavailable beyond a certain time or when the event occurs after the study period (*administrative censoring*), is a common issue. Let $T$ be the time-to-event and $C$ the time-to-censoring, with observed outcome $Y = min(T, C)$ and censoring indicator $\delta = \mathbb{1}(T \leq C)$, where $\delta = 1$ indicates the event was observed. We aim to estimate survival models, focusing on the hazard function, $h(\tau|a) = P(T = \tau|T \geq \tau, A = a)$, and the survival function, $S(\tau|a) = P(T > \tau|A = a)$, which can be used to compute treatment effects. Clinical trials often report the *hazard ratio*, a controversial metric for comparing treatments (Hernán, 2010; Stensrud and Hernán, 2020).

**Estimands of interest.** Survival models characterize the event processes leading to observed time-to-event outcomes. Survival methods center on estimating one of the possible survival models shown in Table 1 (details in App. A.5). In discrete-time, the hazard function is the probability that the individual will experience the event outcome in a given interval of time, where $\tau = [t_{\tau-1}, t_\tau]$. In

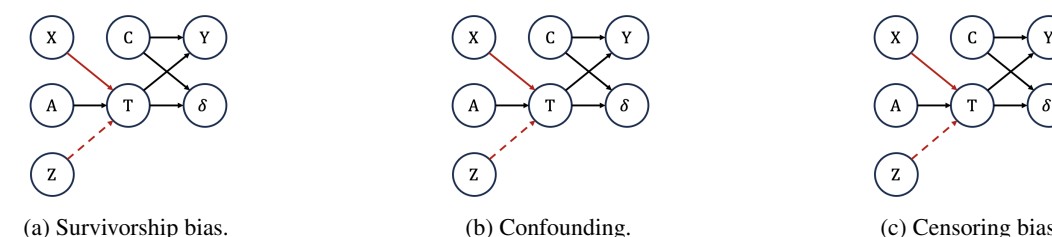

(a) Survivorship bias.  (b) Confounding.  (c) Censoring bias.

Figure 1: Causal diagrams showing biases in the time-to-event setting.

continuous-time, the hazard function is the instantaneous event rate at time $t$ conditioned on survival until $t$. Under causal identifiability conditions (Sec. A.4), the *causal* treatment-specific conditional survival and hazard functions are equivalent to the treatment-specific conditional survival and hazard functions, such that $h^a(\tau|x) = h(\tau|a, x)$ and $S^a(\tau|x) = S(\tau|a, x)$ (derivation in App. A.6). These causal quantities can be used to directly compute treatment effects. While population-level (*average*) treatment effects (ATEs) are typically reported in clinical research (Faraone, 2008), interest in ML methods for estimating heterogeneous (*conditional average*) treatment effects (HTEs) has grown (Alaa and van der Schaar, 2018; Chapfuwa et al., 2021; Cui et al., 2020). Treatment effects are also represented by contrasts between the causal treatment-specific survival and hazard functions, which illustrate the relative benefit of one treatment over another. Clinical trials often report ATEs using the *hazard ratio* (HR), comparing the treatment ($a = 1$) to a control ($a = 0$), as $HR(\tau) = \frac{h^1(\tau)}{h^0(\tau)}$. The marginal HR is controversial in its causal interpretation, especially when treatment effects vary over time (Hernán, 2010; Hernan and Robins, 2023; Martinussen et al., 2020). While use of the conditional hazard ratio, $HR(\tau|x)$, may resolve issues of causal interpretation A.7, it is difficult to estimate. Researchers have encourage use of alternative effect measures (App. A.8).

## 3 CHALLENGES: ESTIMATING TREATMENT EFFECTS FROM SURVIVAL DATA

**Heterogeneity and effect modification.** While RCTs typically report population-level ATEs, these can be misleading when treatment effects vary across different values of covariates $X$, known as *effect modifiers*. In such cases, HTEs are more informative; relying solely on ATEs is problematic, especially for health equity. For example, women and people of color have been historically under-represented in clinical trials, likely leading to biased ATE estimates that do not reflect diverse populations (Chien et al., 2022). As covariates are likely to influence treatment effects, HTEs are a more useful and fair quantity to focus upon, particularly as developments in ML make HTE estimation feasible. Treatment effects can also vary over *time*, complicating estimation further.

**Conditions for estimation of causal treatment effects.** Clinical research aims to determine the *causal effects* of treatments on health outcomes, reflecting how treatments would impact the same population in a counterfactual world. Because multiple treatments cannot be applied to the same individuals, estimating causal effects from observed data requires adherence to *identifiability* conditions. Time-to-event data introduces additional challenges due to censoring and also relies on *exchangeability*, the assumption that the counterfactual risk in the treated population is the same as in the entire population if everyone were treated. If *conditional exchangeability* holds, conditional causal effects can be estimated via methods like inverse propensity weighting (Hernan and Robins, 2023). We depict causal graphs of the time-to-event problem setting in Fig. 1, adapted from related work (Nagpal et al., 2022). See detailed discussion of causal identifiability in App. A.4.

**Confounding and selection bias.** Observational data may be subject to *confounding* (Fig. 1b), where treatment effects are obscured by a common cause of both treatment assignment and patient outcome. Both RCT and observational data may be subject to *selection bias*, where the analyzed population is selected on a common cause (or effect) of both treatment and outcome (Hernan and Robins, 2023). *Censoring* can be a form of selection bias, if the censoring mechanism depends on a covariate that affects treatment outcomes. Confounding and selection bias complicate treatment effects estimation by disrupting *exchangeability* and may also result in *covariate shifts* (Curth et al., 2021a) in the analyzed population, biasing model estimates.

**Covariate shift.** While lack of exchangeability leads to invalid causal interpretation, covariate shift leads to bias during model estimation, particularly in the presence of model misspecification (Shi-

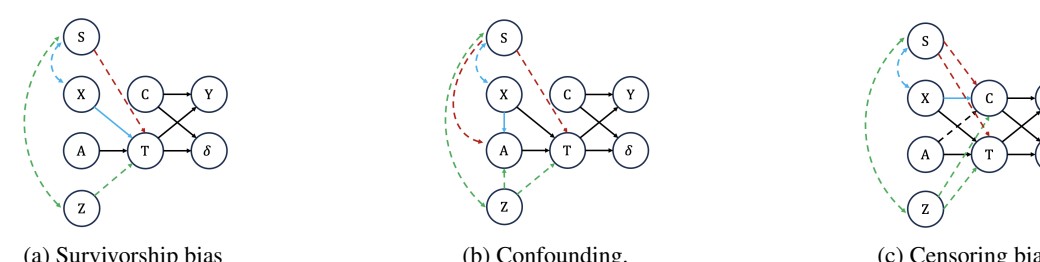

(a) Survivorship bias            (b) Confounding.            (c) Censoring bias.

Figure 2: Biases with subgroup interaction. **Red**: $S$ directly affects bias. **Blue**: $S$ indirectly affects bias via covariate shift on $X$. **Green**: $S$ indirectly affects bias via outcome shift on $Y$ (through covariate shift on unmeasured variables $Z$.) As in (Pfohl et al., 2023), bi-directional arrows indicate that $S$ affects the distribution of the other variable, **not** that it is a direct cause.

modaira, 2000). In this case, a model trained through expected risk minimization (ERM) of the training distribution will be biased with respect to the test distribution (Gretton et al., 2009). This occurs as trained models may fit the data well in regions where $\mathbb{P}_{train}(X)$ is of high probability, but not where $\mathbb{P}_{test}(X)$ is of high probability (Scholkopf et al., 2012).

**Survivorship bias.** A form of selection bias termed *survivorship bias*, also known as "the built-in selection bias of hazard ratios" (Hernan and Robins, 2023), has been discussed extensively in the survival literature (Hernán, 2010; Martinussen, 2021; Martinussen et al., 2020; Stensrud and Hernán, 2020). It occurs in both RCTs and observational data, regardless of study design. Simply stated, if treatment affects outcomes (Fig. 1a), the treatment-specific surviving populations diverge from the baseline and from each other, breaking exchangeability (see App. A.7). While the conditional hazard ratio is causal if the potential outcomes are independent conditional on measured covariates, $T^0 \perp\!\!\!\perp T^1 | X$ (essentially, all effect modifiers, confounders, and sources of selection bias are observed), this is both untestable and unlikely (Martinussen, 2021). Fig. 7 illustrates how unmeasured covariates $Z$ undermine that causal interpretation of the HR. Despite these issues, the HR remains standard in clinical trials, while researchers continue to investigate methods for causally interpretable estimation of HRs (Axelrod and Nevo, 2022; Adib et al., 2020).

**Health equity.** In survival analysis for clinical research, health equity concerns often arise across subgroups defined by *protected attributes*, such as race, ethnicity, or gender. Due to space constraints, the relevant figure (Fig. 2) can be found in App. A.3. Fig. 2 highlights three ways subgroup membership $S$ can drive inequities. 1) Directly, where $S$ affects both survival times and the mechanism of bias (red arrows): this could lead to inequities via the strength of the impact (i.e., subgroup is strongly associated with assignment to a certain treatment) or subgroup prevalence in the data. 2) Indirectly, via covariate shift (blue arrows): patient covariates $X$ and subgroup $S$ may be dependent, such that the distribution of $X$ differs across subgroups, while $\mathbb{P}(Y|X)$, the outcome distribution conditional on covariates, remains the same. Model performance can degrade for covariates that are underrepresented in the training data, due to the presence of 'harder' examples of $X$ or infrequent/unseen values of $X$ (Cai et al., 2023) and exacerbated by small datasets or mis-specified models (Shimodaira, 2000). Because certain demographic groups have been historically underrepresented in clinical trials, unbalanced subgroup distributions leading to model bias is a significant equity issue. 3) Indirectly, where $S$ affects the distribution of unmeasured variables $Z$ (green arrows). This causes *outcome shift*, such that $\mathbb{P}(Y|X)$ differs across subgroups and conditional exchangeability no longer holds. This can exacerbate health inequity if the frailties $Z$ are dependent on protected attributes $S$.

## 4   BENCHMARKING OF SURVIVAL ESTIMANDS

We aim to evaluate (1) survival models and (2) treatment effects contrasts. Since true survival functions, hazard functions, and counterfactual outcomes are unknowable from observed data, which embeds confounding and selection bias, we generate treatment assignments and survival times to simulate various realistic scenarios. This allows us to assess methods under different conditions, such as observational biases and modelling assumption violations. The pseudocode for generating semi-synthetic evaluation data can be found in Alg 1 (App. B.1).

### 4.1 CONSTRUCTING AN EVALUATION SCENARIO

**Discrete vs. continuous time.** Survival times drawn from either discrete-time or continuous-time distributions can be used for the evaluation of both discrete-time and continuous-time methods. However, it is necessary to be mindful of the distinctions. For example, take a scenario where we have generated survival times from a discrete-time model to test a continuous-time method. The resultant continuous-time hazard function must be converted into discrete-time, using Equation 2, to evaluate against a ground truth discrete-time model.

**Data generating components.** Generating synthetic survival data requires the following components: a treatment assignment mechanism, $A \sim Bernoulli(\alpha(x))$, event and censoring processes $h(t|a,x)$ and $h_c(t|a,x)$ (or other survival models from Table 1), which define how frequently an event/censoring event occurs, and patient covariates. Covariates can be generated synthetically or sampled from real datasets to mimic realistic experimental conditions.

**Model mis-specification.** Parametric and semi-parametric methods require the underlying event process to adhere to an assumed form that may not reflect the true data distribution. For example, we can evaluate the CoxPH in situations where the proportional hazards assumption is violated. We can evaluate parametric models, such as the exponential model, against misspecified data generated from a log-logistic distribution. Time-varying models may be mis-specified with respect to the time function. Table 5 summarizes types of mis-specification and what methods are affected.

**Heterogeneous treatment effects**. Treatment effects may be heterogeneous from two perspectives: the event/censoring processes are dependent on covariates or the causal contrasts are dependent on covariates. The former refers to a scenario where $h^a(t|x) \neq h^a(t)$. The latter refers to a scenario where, if we define the causal contrast as the hazard ratio, $HR_A(t|x) \neq HR_A(t)$.

**Time-varying treatment effects.** Similar to heterogeneity, treatment effects may be time-varying from two perspectives: the event/censoring processes are dynamic, or the treatment effects (causal contrasts) are dynamic. For example, survival times drawn from an exponential distribution reflect a constant hazard function, where $h(t) = \lambda$. A time-varying hazard function does not necessarily imply time-varying causal contrasts. For example, if $h^a(t) = \lambda t \exp(0.2a)$, $HR_A(t) = \exp(0.2)$, which is constant over time. A time-varying contrast is $HR_A(t) = \exp(0.1t)$, which would result from $h^a(t) = \exp(0.1a \cdot t)$. This complication can be combined with the above to generate heterogeneous *and* time-varying treatment effects, such that $HR_A(t|x) = \exp(0.1t + 0.2x)$, from $h(t|a,x) = \exp(0.1a \cdot t + 0.2a \cdot x)$, or *heterogeneously* time-varying treatment effects, such that $HR_A(t|x) = \exp(0.1t \cdot x)$, from $h(t|a,x) = \exp(0.1a \cdot t \cdot x)$.

**Observational bias.** Bias can be incorporated into a synthetic scenario via the inclusion of common effects, defined as patient covariates. Confounding occurs when there is a common cause of treatment assignment and patient outcome. For example, if $A \sim Ber(\sigma(x_1))$ and $h^a(t|x) = \exp(a \cdot x_1 + x_2)$. Selection bias occurs when the at-risk population is selected based on two variables: treatment or cause of treatment and outcome or cause of outcome. We can create a censoring mechanism that incorporates selection bias if we make it dependent on variables that satisfy this definition. For example, if treatment assignment and the hazard function are defined as above, the censoring hazard $h_c^a(t|x) = exp(a \cdot x_1)$ leads to censoring bias. Recall that survivorship bias occurs in any situation where there exist covariates (including treatment) that affect survival.

**Violation of identifiability assumptions.** Real world data is likely to violate causal identifiability assumptions. The conditional exchangeability assumption does not hold if there are unmeasured confounders, or, in the presence of censoring, there are any unmeasured variables that affect both censoring and outcome. To create scenarios that violate these assumptions, we can incorporate variables into the data generating models for treatment assignment, event hazard, and censoring hazard that are withheld during model training.

### 4.2 EVALUATION

Calibration (which is related to the notion of *sufficiency* (Barocas et al., 2023)) has often been touted as an appropriate measure of fairness (Pleiss et al., 2017), particularly in healthcare settings, for evaluating models for clinical decision making (Pfohl et al., 2022). Thus, we focus on calibration as our primary evaluation metric. In the context of survival analysis, we say a model is perfectly calibrated if the estimated hazard function equals the true hazard function. It remains to determine

how best to quantify deviations from perfect calibration. Given a time interval $\tau$, a ground truth (discrete) hazard function $h$, and an estimated hazard $\hat{h}$, we define the *absolute logit error (ALE)* as

$$\text{ALE}(\tau) = \left| \log \frac{h(\tau|a,x)}{1 - h(\tau|a,x)} - \log \frac{\hat{h}(\tau|a,x)}{1 - \hat{h}(\tau|a,x)} \right|. \tag{1}$$

The main evaluation metric we use in our experiments is the *mean absolute logit error (MALE)*, which is just the ALE averaged over all failure intervals.

**Motivation for MALE.** MALE has two properties which make it an ideal performance metric. First, it takes its minimum value 0 if and only if the model is perfectly calibrated, i.e., iff the estimated hazard function is equal to the true hazard function. Second, a bound on MALE (which is in terms of hazards) also corresponds to a natural bound on the difference in log failure probabilities between the model and the ground truth. In contrast, while other seemingly natural measures such as the mean squared error of the true vs. estimated hazard share the property that they are minimized only by the ground truth, bounds on these quantities give no guarantees about the relative error of the computed survival probabilities. Formal statements and proofs of these qualities can be found in Appendix D.1.

## 5 EXPERIMENTS

In this section, we benchmark common survival methods on their ability to estimate hazard functions from survival times and patient covariates. We emphasize that we are interested in the estimation of hazard functions (rather than predicting survival times) as they can be used to estimate treatment effect contrasts, such as the hazard ratio, which are informative for clinical decision making. We evaluate models on datasets generated with a wide range of ground truth hazards that address model misspecification, constant vs. time-varying hazards, and confounding, as well as examining subgroup fairness. While thorough, our experiments are non-exhaustive; alongside the insights reported, we hope that this work can help improve evaluation practices ML for survival analysis.

**Datasets.** We use covariates both synthetically generated (App. B.2) and sampled from real datasets. We utilize a diverse set of real-world datasets that vary in sample size, number of features, and feature characteristics, and have been widely used in related work on survival analysis and treatment effects estimation. These include Twins (Almond et al., 2004), TCGA (Weinstein et al., 2013), IHDP (Hill, 2011), News (Johansson et al., 2016), SUPPORT (Connors et al., 1995), and METABRIC (Curtis et al., 2012). The characteristics of the real-world datasets are summarized in Table 2, with more details in App B.3. Because it is not possible to know true underlying hazard or survival functions from observed data, nor is it possible know if treatment assignments are affected by confounding, we assign treatments we generate synthetic survival (and confonding) times using the procedure detailed in Alg 1. Hazard and censoring hazard functions corresponding to scenarios are shown in Table 3. Unless otherwise noted, all scenarios incorporate non-informative censoring.

**Methods.** We evaluate the performance of the parametric logistic hazard model (LH) (Tutz and Schmid, 2016), the semi-parametric Cox proportional hazards model (CoxPH) (Cox, 1972), the time-varying Cox model (CoxTV), Random survival forests (RSF) (Ishwaran et al., 2008), and neural network methods DeepSurv (Katzman et al., 2016), CoxTime (Kvamme et al., 2019), and DeepHit (Lee et al., 2018). See App. B.4 for detailed discussion and implementation details.

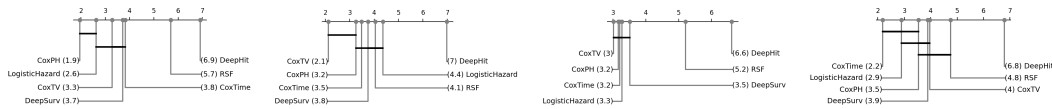

(a) Baseline, constant-time scenarios.    (b) Baseline, time-varying scenarios.    (c) Mis-specified, constant-time scenarios.    (d) Mis-specified, time-varying scenarios.

Figure 3: Critical difference diagrams of average ranks (based on MALE).

**Overall results.** Fig. 3 shows critical difference (CD) diagrams comparing the rankings of survival methods based upon MALE performance over baseline and mis-specified scenarios across all 7 datasets. We first conduct a Friedman test (Friedman, 1937), finding that differences in model performance are statistically significant ($p < 0.05$). Then, we construct CD diagrams using the results

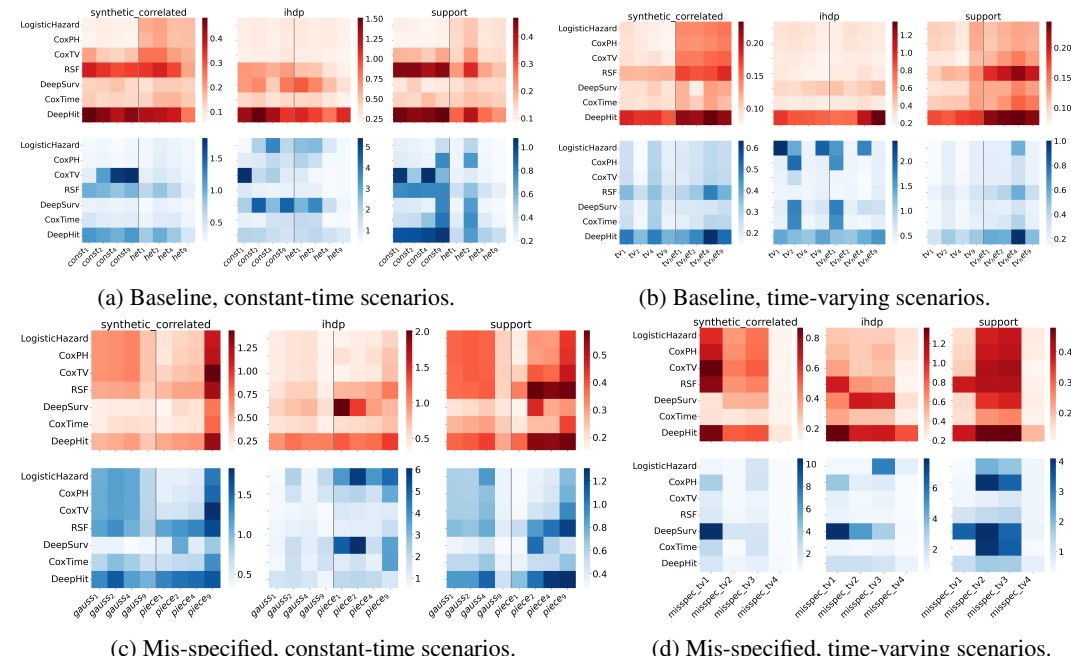

(a) Baseline, constant-time scenarios.

(b) Baseline, time-varying scenarios.

(c) Mis-specified, constant-time scenarios.

(d) Mis-specified, time-varying scenarios.

Figure 4: Heatmaps comparing MALE (heatmap rows) over scenarios (heatmap columns) over datasets (columns). MALE is reported up to the 75th percentile of survival times (top row) and 76th to 99th percentile of survival times (bottom row). Grey lines group variations of similar scenarios. Note that heatmap scales are all different.

of the Nemenyi post-hoc test (Nemenyi, 1963), which determines which models have statistically significant different rankings. We report the rankings based on MALE performance average over the 75th percentile of survival times, due to stark differences in model performance in later survival times (99th percentile rankings in Fig. 8). Model rankings are largely intuitive: for baseline, constant-time scenarios, CoxPH and LH are well-specified and ranked highly; for baseline, time-varying scenarios, CoxTV is well-specified and ranks highly (Fig. 3b); for mis-specified, time-varying scenarios, CoxTime, an extension of CoxTV parameterized by a neural network, is ranked highly (Fig 3a). Strangely, for mis-specified, constant-time scenarios, CoxTV slightly outranks other methods (though without statistical significance), possibly as CoxTV incorporates additional time parameters, which may offer more flexibility in the presence of non-linearities. Notably, across all groups of scenarios, no other models outperform (with statistical significance) the simplest models: LH and CoxPH. Results are discussed in more detail in Secs. 5.1 and 5.2.

## 5.1 BASELINE SCENARIOS

**Setup.** We examine model performance over a set of baseline scenarios with constant-time and time-varying, heterogeneous hazard functions, without the additional complications. While parametric/semi-parametric LH and CoxPH models are technically mis-specified to heterogeneous hazard ratios and time-varying hazards, these scenarios represent a baseline over which survival models should perform reasonably well. While we aim to examine a breadth of scenarios representing different manifestations of heterogeneity and time effects, we are also interested in varying the complexity of hazard functions. $\mathcal{I}_{haz} \subseteq |d|$ represents the indices for the covariates that affects the hazard function. We increase $|\mathcal{I}_{haz}|$ over variations of similar scenarios (exact hazard functions shown in Table 3); the value of $|\mathcal{I}_{haz}|$ for each scenario is included as its subscript. For example, $const_1$ includes 1 covariate that affects hazard: $h(t|a,x) = 0.5 * \exp(-2 + a + x_0)$. For experiments on real datasets, covariates are indexed randomly from the set of available covariates.

**Results.** Figs. 4a and 4b show MALEs of models (heatmap row) over the baseline scenarios (heatmap column), for a select subset of datasets (columns). Results for remaining datasets are in App. Fig. 9. For each dataset, MALEs are average over survival times up to the 75th percentile (top row) and

from the 76th to 99th percentile (bottom row). As the risk set shrinks at later survival times, model performance can deteriorate quickly, obscuring performance patterns when metrics are averaged over time. For constant-time scenarios (Fig. 4a) LH and CoxPH generally perform well and performance does not typically deteriorate for later survival times, except for IHDP, which is the smallest dataset ($n = 985$). CoxTV deteriorates significantly in later survival times for some constant-time scenarios, even with the larger synthetic dataset ($n = 10000$); this performance decline is not seen in the time-varying scenarios (Fig. 4b). This is possible a drawback to use of CoxTV–if practitioners are unaware of whether the ground truth hazard is time-varying, CoxTV may perform poorly. Interestingly, the performance deterioration that is observed in DeepSurv in IHDP is not observed in CoxTime, a similar neural network that includes time as a parameter. While the LH and CoxPH are mis-specified to time-varying scenarios, in practice they perform quite well, with the exception of some scenarios of IHDP, where performance deteriorates significantly at later survival times.

## 5.2 MISSPECIFICATION: NON-LINEARITIES

**Setup.** We now explore model performance over mis-specified hazard functions, particularly, non-linearities, which are mis-specified to the parametric LH model and the semi-parametric CoxPH and CoxTV models. Other types of mis-specification, including time-varying hazards (Section 5.1) and unmeasured variables (Sec. 5.3) are explored in other sections. Scenarios in Figure 10a are constant over time, while scenarios in Figure 10b are time-varying, with non-linearities over time as well as the covariate space. Time-varying, non-linear hazard functions are mis-specified to DeepSurv, which does not model time-varying covariates. In $gauss$ scenarios, the hazard function mimics a Gaussian distribution, while in $piece$, the hazard function is a linear piecewise function. The time-varying non-linear functions are more variable, full details are found in Table 3.

**Results.** We report average MALE values averaged across time periods for mis-specified, constant-time and mis-specified, time-varying scenarios for a select subset of datasets in Fig. **??**. Heatmaps for the rest of the datasets can be seen in Fig. 10. While the reported model rankings in Fig. 3 suggest otherwise, we find that generally, DeepSurv outperforms other methods (across all time periods) in mis-specified, constant-time scenarios (Fig. 10a. Average rankings are skewed particularly by results on the dataset *METABRIC*. If *METABRIC* results are removed, DeepSurv ranks first on average (though still not statistically significantly). For mis-specified, time-varying scenarios (Fig. 10b), CoxTime generally performs the best (this is reflected in average model rankings as well); DeepSurv notably performs quite poorly in small data settings (IHDP) and also at later survival times (bottom row). As practitioners may not know ahead of time whether a hazard will be time-varying (as well as non-linear), it may be advisable to select CoxTime rather than DeepSurv; CoxTime generally performs well even in constant-time scenarios.

## 5.3 OBSERVATIONAL BIAS: CONFOUNDING

**No unmeasured confounders.** We examine the impact of confounding, a bias that occurs when there exists covariates that affect both treatment assignment and patient outcomes. In a setting where there are no unmeasured confounders, biases caused by confounding can be accounted for by the estimation of conditional hazards (conditioning on any confounders). However, confounding may still lead to covariate shift that can result in model estimation biases in low data or mis-specified settings. We are interested in investigating what impacts the strength of confounding and the complexity of the confounding mechanism have upon model performance. $\mathcal{I}_{haz} \subseteq |d|$ represents the indices for the covariates that affect the hazard function. $\mathcal{I}_{con} \subseteq \mathcal{I}_{hazard}$ represents the set of indices for covariates that also affect treatment assignment. Here, the hazard function can be written as $h(\tau|a, x) = h(\tau|a, x_{\mathcal{I}_{haz}})$. We define a treatment assignment function with a weight parameter, $\omega_c$, which controls confounding strength: $\alpha(x) = \sigma(\omega_c \cdot \sum_{j \in \mathcal{I}_{con}} x_j)$.

**Results.** In Figure 5a, we depict experimental results over varying the number of covariates that affect treatment assignment, $|\mathcal{I}_{con}|$, shown across the figure rows, and also varying the strength of the confounding, shown over the y-axis of each plot. Figure columns correspond to assigned treatment groups, $a$. In this experiment, we use hazard function $piece_9$, where $|\mathcal{I}_{haz}| = 9$. Higher values of $\omega_c$ correspond to starker differences in the distributions of the covariates within $\mathcal{I}_{con}$ by treatment assignment. The increasing size of $\mathcal{I}_{con}$ means that more covariates (which also affect hazard) will be affected by distribution shift between treatment groups. We observe that generally, as $|\mathcal{I}_{con}|$

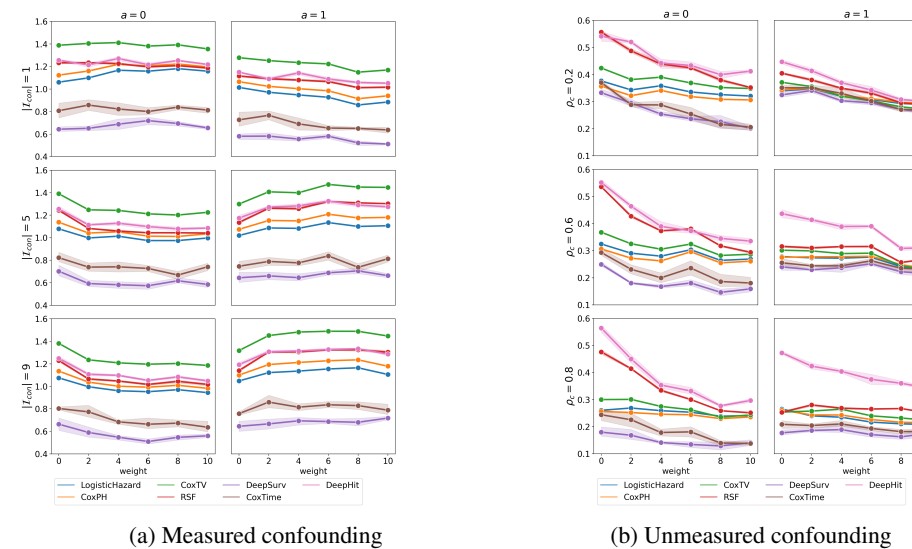

(a) Measured confounding  (b) Unmeasured confounding

Figure 5: Observational bias: confounding

increases, performance across all models seems to slightly improve for treatment 0, but worsen for treatment 1. Due to the construction of $\alpha(x)$, as $\omega_c$ increases, the distribution of covariates (which affect confounding) for treatment 1 tend towards larger values, while the distribution of covariates for treatment 0 tend towards smaller values. In the hazard function used for this experiment, smaller values (across multiple dimensions) tend towards lower hazard probabilities. The hazard function also incorporates a non-linearity at higher covariate values. Thus, the structure of both the confounding mechanism and the underlying hazard function could explain these changes in performance. For higher $|\mathcal{I}_{con}|$ values (rightmost columns), we see that changes in $\omega_c$ have the largest impact on performance. These performance can also be explained by the pattern of distribution shift–treatment group 0 is increasingly associated with an 'easier' (well-specified, low hazard probabilities) covariate space, while the opposite is true of treatment group 1. While these results are specific to the experimental conditions, we demonstrate that confounding, even when all confounders are observed, can influence model performance across all investigated models. We also see that ranking of model performance remains fairly consistent, with neural network methods, DeepSurv and CoxTime, which are able to address the non-linearities in hazard function $piece_9$, outperforming other methods.

**Unmeasured confounders.** We turn our attention to investigate model behavior in the face of unmeasured confounding. In this setting, we have unobserved confounders $z$ which also affect both hazard and treatment assignment. Here, the hazard function is $h(\tau|a, x, z)$ and the treatment assignment function is $\alpha(z) = \sigma(\omega_c \cdot z)$ (in these experiments, $z$ is 1-dimensional). Here, we are interested in what happens if we increase the confounding strength $\omega_c$ and also vary the correlation strength, $\rho_c$, between $z$ and $x$. Because $z$ is unmeasured and thus not included in model estimation, the correlation between $z$ and observed variables $x$ should impact model performance. In this experiment, we use hazard function $piece_4$, which is constant-time and incorporates non-linearities.

**Results.** In Fig. 5b we show model performance over increasing correlation strength between $z$ and the observed variables $x$ (columns) and increasing confounding strength (y-axis). As expected, performance generally improves across all models as the correlation strength increases (with the exception of DeepHit). Interestingly, across all correlation strengths and models, performance seems to improve in both treatment groups as $\omega_c$ increases. A possible explanation for this phenomenon is that the increased confounding strength leads to treatment group distributions that are concentrated in separate covariate spaces that may be 'easier' for the models to learn. This is indeed the case for hazard function $piece_4$, a non-linear piecewise function where the breakpoints occur around the middle of the covariate space. The covariate shifts caused by the unmeasured confounding appear to affect the parametric and semi-parametric LH and CoxPH models the least, likely as these models are not flexible enough to adapt to the non-linearities, regardless of covariate distribution.

## 5.4 SUBGROUP FAIRNESS: SURVIVORSHIP BIAS

In this section, we investigate one possible source of inequity in the survival setting: survivorship bias linked to protected attributes, $s$, which indirectly affect bias via covariate shift on measured covariates, $x$. In the appendix, we also report results on an experiment investigating inequity caused by covariate shift on unmeasured covariates, $z$, which are affected by subgroups $s$.

**Setup.** We now examine a setting where participants may belong to subgroups defined by protected attributes, $s$. Even when there are no observational biases (such as confounding or informative censoring) and all covariates that affect treatment outcome are measured, we still contend with the issue of survivorship bias. Survivorship bias may be a source of inequity when, for example, subgroup membership indirectly affects this bias via covariate shift on $X$, such that $\mathbb{P}(X|S)$. Here, the hazard function $h(\tau|a, x, s)$ is complicated further by the presence of subgroups. In our experimental setup, we draw $x_0$ based on subgroup membership $s$, where $x_0|s = 0 \sim \mathcal{N}(\mu_0, \sigma^2)$ and $x_1|s = 1 \sim \mathcal{N}(\mu_1, \sigma^2)$. We vary subgroup means $\mu_0$ and $\mu_1$ across experiments (by column) to investigate impacts on model performance as the overlap between subgroups decreases. We also vary the proportion of subgroups $P(s = 1) = \pi$ (shown on y-axis). We use hazard function $piece_1$, which is constant-time, non-linear, piecewise function where $\mathcal{I}_{haz} = \{0\}$ ($x_0$ is the only covariate that affects the hazard). We report MALE values, averaged over the first 75th percentile of survival times.

**Results.** We report findings in Figure 11 which can be found in Appendix C.3 due to space constraints. Each row of the figure shows an experiment with different subgroup means, $(\mu_0, \mu_1)$, and each column is associated with a treatment $a$ and subgroup $s$ pair. When the subgroup distributions are closer together (first row), model performance generally remains similar across treatment, subgroup pairs, with the exception of DeepHit and RSF, where performance is significantly worse for $(a = 1, s = 0)$ and $(a = 0, s = 1)$. We find that this disparity increases as the subgroup means get further apart (lower rows). In $(a = 1, s = 0)$, model performance worsens as $\pi$ increases, which means that fewer samples belong to subgroup 0; as a result, the model errs in this covariate space. We see the reverse effect in the columns corresponding to subgroup 1, particularly when $a = 0$; performance improves $\pi$ increases and more samples represent subgroup 1. These effects occur at a smaller scale for other methods as well; as $\pi$ increases, performance degrades for those in subgroup 0 and improves for those in subgroup 0. Stark differences in performance created by changes in subgroup distribution terms of of both sample size and covariate shift) across treatment, subgroup pairs are indicative of unfairness. Thus, even while all effect modifiers are observed, survivorship bias may have a significant bias on model performance, particularly for subgroups that are not well-represented in the dataset. This effect is seen most clearly in DeepHit and RSF and to some degree with CoxTV as well.

## 6 DISCUSSION

We have provided a comprehensive discussion of the time-to-event problem setting, alongside the requisite assumptions for causal inference of treatment effects and significant challenges faced during model design and estimation. We provide recommendations for benchmarking and evaluating methods for time-to-event treatment effect estimation, and evaluate the Cox proportional hazards model and the extended Cox model under this paradigm. In these experiments, we expose the common biases that the Cox model exhibits under different event settings, finding that the it is not robust to model misspecification (either due to the parameterization of the distribution or unmeasured confounders/effect modifiers) and is easily biased by certain covariate shifts. Future work should focus on further examining the Cox model with respect to these issues and towards the development of methods to overcome these particular forms of covariate shift.

This work also has several limitations. Most notably, we consider a specific setting for time-varying treatment effects, wherein the treatment is set at the beginning of analysis and covariates are measured at the beginning of analysis (but may contribute to time-varying effect modification). However, clinical data may actually include time-varying covariates, time-varying treatments, and multiple treatments. In addition, patients may be subject to competing events, where the patient outcomes are caused by multiple mechanisms. There are also many more forms of selection bias, including selection that occurs before a trial/before analysis, affecting generalizability of findings from analysis to the general population. For example, eligibility criteria from clinical trials may restrict the investigated population to a subset that is not reflective of the eventual patient population. This is a form of selection bias also known as *sampling bias* and affects the transportability of clinical trials outcomes.

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

# A    ADDITIONAL BACKGROUND

## A.1    CURRENT PRACTICES

### A.1.1    THE COX PROPORTIONAL HAZARDS MODEL

We discuss the *Cox proportional hazards model (Cox PH)* (Cox, 1972) with more detail, due to its popularity. The Cox PH is a semi-parametric approach to modelling hazard functions in continuous-time, with the primary goal of calculating hazard ratios. For estimation of average treatment effects, the hazard function for the Cox PH is defined as $h(t|a) = h_0(t) \exp(\beta A)$. Here, the baseline hazard $h_0(t)$ is assumed to be the same across treatment groups and differs only by a constant (over time) scaling factor, $exp(\beta A)$, which is dependent on the treatment assignment. This is the *proportional hazards assumption*, which is violated if the treatment effect coefficients $\beta$ vary over time. The Cox PH model is estimated with a partial likelihood (Cox, 1975) that relies on the *censoring at random* assumption and treats the baseline hazard, $h_0(t)$, as a nuisance parameter that is not required in the likelihood definition and not estimated during model inference. However, if a baseline survival model is desired, the Breslow estimator is commonly employed to estimate the cumulative baseline hazard (Lin, 2007). The Cox PH model is useful in that it does not require any assumptions regarding the parametric form of the baseline hazard and the common issue of right-censoring is handled in model inference. Additionally, the Cox PH presents a straightforward estimation of the hazard ratio, which is often desired as a metric for the comparison of treatment effects. The Cox PH marginal hazard ratio is defined as: $HR(t) = \frac{h(t|A=1)}{h(t|A=0)} = \frac{h_0(t) \exp(\beta*1)}{h_0(t) exp(\beta*0)} = exp(\beta)$.

However, there are two major issues with this definition of the hazard ratio: 1) dependence on the proportional hazards assumption, and 2) survivorship bias. Recalling the definition of the hazard ratio from Equation 6, we note that the hazard ratio is meant to be interpreted as a contrast between treatment groups of the probability of event at a specific moment in time, $t$, conditioned on survival until that time. However, the hazard ratio given by the Cox PH model really reflects a *weighted average* of the hazard ratios over the entire time period of $0 \leq u \leq t$ (Stensrud and Hernán, 2020), rather than the hazard ratio at the specific moment in time, $t$. Researchers have proposed to resolve this by reporting a series of period-specific hazard ratios in order to reflect time-varying hazards (Lin et al., 2019). This is problematic due to issue (2) noted in the previous paragraph; if treatment-specific event processes differ, the distributions of the at-risk treatment groups diverge over time and lack exchangeability (Bartlett et al., 2020). The conditional Cox model follows by incorporating patient covariates $X$ in the model in the same format as the treatment assignment variable, $A$. Under strict assumptions (Martinussen, 2021), the conditional HR can be interpreted causally.

**The extended Cox model.** If the proportional hazards assumption is violated due to the presence of time-varying treatment effects, the *extended Cox model* can be adopted (Bao et al., 2018) to model time-varying covariates or time-varying coefficients. We focus on the modelling of time-varying coefficients, indicating varying treatment effects influenced by effect modifiers. Under the extended Cox model, the hazard function is modified such that $h(t|a) = h_0(t)\exp(\beta(t) \cdot A)$, where the coefficients vary over time. If the time-varying coefficient can be represented by a time function, $g(t)$, such that $\beta(t) = \beta g(t)$, then the Cox model can be used with a set of time-varying variables (Thomas and Reyes, 2014), as $\beta \cdot g(t) \cdot A = \beta \cdot A(t)$. However, this procedures requires an assumption of the time function, $g(t)$, which provides another opportunity for model specification.

### A.2 GENERAL CAUSAL MODEL

The causal setting of Nagpal et al. (2022). We incorporate treatments $A$ and unmeasured factors $Z$.

### A.3 CAUSAL MODELS FOR HEALTH EQUITY

### A.4 IDENTIFIABILITY CONDITIONS

In the following, we use $T^a$ to denote the potential outcomes, the event time that would have been observed given the assigned treatment $a$:

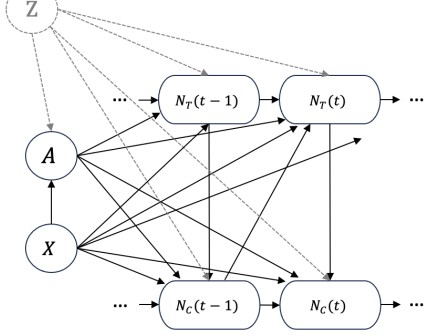

- **Assumption 1** (Consistency). *The observed outcome is the counterfactual outcome under the intervention actually observed. Thus, if $A = a$, then $T = T^a$.*

- **Assumption 2** (Conditional exchangeability). *The counterfactual outcome and the assigned treatment are independent conditional on measured covariates, such that $T^a \perp\!\!\!\perp A|X$. Conditional exchangeability requires the presence of no unmeasured confounders, where all variables that affect both treatment assignment and outcomes are observed. It can be achieved if treatment assignment is random conditional on measured covariates..*

Figure 6: Causal model of time-to-event setting. $N_T$ represents the event process, $N_C$ represents the censoring event process. $Z$ represents unmeasured variables that can perform effect modification, and cause confounding or selection bias.

- **Assumption 3** (Positivity). *There is a positive probability of treatment assignment to each treatment conditional on patient covariates, such that $P(A = a|X = x) > 0$ for $a \in \{0, 1\}$ and $x$ where $p(x) > 0$, where $p(\cdot)$ is the probability mass function. Positivity is also known as the experimental treatment assumption.*

*Consistency* is required because treatments must be well-defined in order to then estimate their causal effects (Rubin, 1980; 1986). Because it is not possible to administer the same treatments to the same individuals to determine the impact of a single treatment, estimation of causal treatment effects relies on the concept of *exchangeability* (Robins and Hernán, 2008). Exchangeability holds when the counterfactual outcome and the assigned treatment are independent, such that the counterfactual risk (of some health outcome) in the treated population is the same as the counterfactual risk (of some health outcome) in the entire population, had the entire population been treated. However, as the risk of treatment is actually observed in the treated population, it can be held true across the entire population (Hernan and Robins, 2023). It is sufficient to require *conditional exchangeability* if there are *no unmeasured confounders*, as methods such as *inverse propensity weighting* can be used to adjust for confounders to estimate average causal effects (Robins and Hernán, 2008). *Positivity* is required as the causal effects of a treatment on patients can only be assessed if representative patients have received the treatment (Hernan and Robins, 2023). These conditions motivate the design of RCTs, where, in ideal settings, all identifiability conditions are achieved by construction (Hernan and Robins, 2023). Consistency is achieved through complete adherence to the assigned treatment protocol, which is carefully designed. Exchangeability and positivity are achieved via randomization of treatment assignments. While these assumptions are *untestable* for observational data, leading to the presence of unmeasured variables ($Z$ in Figure 7), practitioners can use expert knowledge and careful problem framing in order to improve plausibility.

### A.4.1 IDENTIFIABILITY IN THE PRESENCE OF CENSORING

In the time-to-event setting, patient observations can be *censored* and unavailable after a certain time. In order to uphold the conditions of *identifiability* when censoring is present, additional assumptions are required, which are standard in the field of survival analysis. Assumption 4 is analogous to Assumption 2 of conditional exchangeability, while Assumption 5 is analogous to Assumption 3 of positivity.:

- **Assumption 4** (Coarsening at random / censoring at random). *Censoring and outcome are conditionally independent given assigned treatment and patient covariates, such that* $T^a \perp\!\!\!\perp C | A, X$.

- **Assumption 5** (Positivity (censoring)). *Censoring is non-deterministic, such that for all values of covariates X, there is a positive probability of being uncensored.* $P(C > \tau | A = a, X = x) > 0$, *for all* $\tau < t$.

## A.5 FURTHER DETAILS ON SURVIVAL MODELS

Survival models are defined in Table 1. These estimands can be derived from one another, below, we list the relationships.

Table 1: Survival models in discrete- and continuous-time.

| Model | Discrete Time | Continuous Time |
|---|---|---|
| Survival function | $S(\tau|a) = P(T > \tau|A = a)$ | $S(t|a) = P(T > t|A = a)$ |
| Hazard function | $h(\tau|a) = P(T = \tau|T \geq \tau, A = a)$ | $h(t|a) = \lim_{dt \to 0} \frac{P(t \leq T < t+dt)}{dt \cdot S(t|a)}$ |
| Cumulative hazard function | $H(\tau|a) = \sum_{u \leq \tau} h(u|a)$ | $H(t|a) = \int_0^t h(u|a)\, du$ |
| PMF / PDF | $f(\tau|a) = P(T = \tau|A = a)$ | $f(t|a) = P(T = t|A = a)$ |

### A.5.1 DISCRETE-TIME

- *Survival function:* $S(\tau|a) = P(T > \tau|A = a) = 1 - F(\tau|a) = \exp(-H(\tau|a)) = \prod_{u \leq \tau}(1 - h(u|a))$

- *Hazard function:* $h(\tau|a) = P(T = \tau|\tau \geq t) = \frac{f(\tau|a)}{S(\tau-1|a)} = H(\tau) - H(\tau - 1)$

- *Cumulative hazard function:* $H(\tau|a) = \sum_{u \leq \tau} h(u|a)$

- *PMF:* $f(\tau|a) = P(T = \tau|A = a) = h(\tau|a)S(\tau - 1|a)$

- *Lifetime distribution function:* $F(\tau|a) = P(T \leq \tau) = 1 - S(\tau|a)$

### A.5.2 CONTINUOUS-TIME

- *Survival function:* $S(t|a) = P(T > t|A = a)$

- *Hazard function:* $h(t|a) = \lim_{dt \to 0} \frac{P(t \leq T < t+dt)}{dt \cdot S(t|a)} = \frac{f(t|a)}{S(t|a)}$

- *Cumulative hazard function:* $H(t|a) = \int_0^t h(u|a)\,du = -\log(S(t|a))$

- *PDF:* $f(t|a) = P(T = t|A = a) = F'(t|a)$

- *Lifetime distribution function:* $F(t|a) = P(T \leq t) = 1 - \exp(H(t|a))$

The distinction between discrete- and continuous-time models is particularly important when we wish to use the hazard functions to determine causal contrasts under various data generating settings. Note that previously expressed definitions assumed discrete-time. The discrete-time hazard function, $h_\tau$, can be derived from the continuous-time hazard function, $h_t$:

$$h_\tau(\tau|a) = 1 - \exp\left(-\int_{t_{\tau-1}}^{t_\tau} h_t(u|a)du\right) \tag{2}$$

### A.5.3 IMPACT ON EXCHANGEABILITY

While either confounding or selection bias can lead to a lack of *exchangeability*, if *conditional exchangeability* holds, it is possible to estimate conditional causal effects and to recover exchangeability via methods such as standardization or inverse propensity weighting (Hernan and Robins, 2023). Conditional exchangeability is achieved in the presence of confounding if there are *no unmeasured confounders*, such that any covariate that affects both treatment assignment and outcome is measured and accounted for, and in the presence of selection bias, if there are no unmeasured covariates that affect both the selection mechanism and outcome. For example, when selection occurs through censoring, conditional exchangeability is achieved if all covariates that affects the censoring mechanism are measured (see Assumption 4, Section A.4).

### A.6 DERIVATION OF CAUSAL SURVIVAL MODELS

First, we account for the presence of censoring in the conditional hazard function:

$$
\begin{aligned}
h(t|a,x) &= P(T = t|T \geq t, A = a, X = x) \\
&= P(Y = t, \delta = 1|Y \geq t, A = a, X = x) \\
&= P(T = t|T \geq t, C \geq t, A = a, X = x)
\end{aligned}
\tag{3}
$$

Line one follows by definition of the discrete-time conditional hazard function. Line two follows from Assumption 4, as the conditional probability of hazard given the full dataset with no censoring should be equivalent to the conditional probability of hazard given the observed data of a censored-at-random dataset. This assumption is commonly adopted for in likelihood-based estimation of models from survival data, due to the presence of censoring. Line three follows by definition. We can then employ causal operators to define a *causal* treatment-specific conditional hazard function which is equivalent to the treatment-specific conditional hazard function:

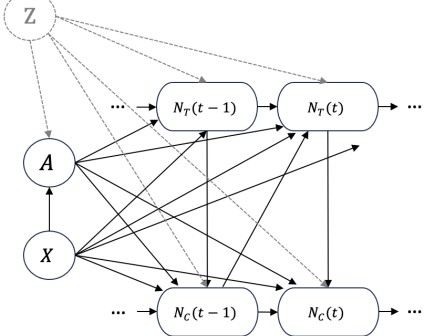

Figure 7: Causal model of time-to-event setting. $N_T$ represents the event process, $N_C$ represents the censoring event process. Z represents unmeasured variables that can perform effect modification, and cause confounding or selection bias.

$$
\begin{aligned}
h(t|a, x) &= P(T = t | T \geq t, C \geq t, A = a, X = x) \\
&= P(T^a = t | T^a \geq t, C \geq t, A = a, X = x) \\
&= P(T^a = t | T^a \geq t, C \geq t, X = x) \\
&= P(T^a = t | T^a \geq t, do(C \geq t), X = x) \\
&= P(T = t | T \geq t, do(A = a, C \geq t), X = x) \\
&= h^a(t|x)
\end{aligned}
\tag{4}
$$

The first line follows from definition, line two follows from Assumption 1 (consistency), line three follows from Assumption 2 (conditional exchangeability), and line four follows from Assumption 4 (censoring at random). The final line follows from definition and gives us the our formula for a causal, treatment-specific hazard function, $h^a(t|x)$. As we are working with time-to-event data, it is necessary for us to intervene on the censoring mechanism (by setting each individual to uncensored), so that we can evaluate treatment outcomes as if we exist in a world without censoring (Stitelman and van der Laan, 2010). The causal, treatment-specific survival function can be defined similarly, as:

$$
\begin{aligned}
S^a(t|x) &= P(T > t | do(A = a, C \geq t), X = x) \\
&= P(T_a > t | C \geq t, X = x)
\end{aligned}
\tag{5}
$$

## A.7 Causal hazard ratios

We define the marginal hazard ratio:

$$
HR(\tau) = \frac{P(T^1 = \tau | T^1 \geq \tau, C \geq \tau)}{P(T^0 = \tau | T^0 \geq \tau, C \geq \tau)}
\tag{6}
$$

The marginal HR compares the surviving (and uncensored) treated population $\mathbb{P}(T^1 \geq \tau, C \geq \tau)$ with the surviving (and uncensored) control population, $\mathbb{P}(T^0 \geq \tau, C \geq \tau)$. If the treatments indeed have different effects on the outcome, these two groups are no longer exchangeable and the resultant HR cannot be regarded as a causal effect. To resolve this issue, Martinussen et al. (2020) introduces a *causal hazard ratio* over the population-average, defined as:

$$
HR_C(\tau) = \frac{P(T^1 = \tau | T^0 \geq \tau, T^1 \geq \tau, C \geq \tau)}{P(T^0 = \tau | T^0 \geq \tau, T^1 \geq \tau, C \geq \tau)}
\tag{7}
$$

The causal hazard ratio constructs an exchangeable population, $\mathbb{P}(T^0 \geq \tau, T^1 \geq \tau, C \geq \tau)$, so that estimand now represents a valid causal contrast. However, $HR_C(\tau)$ can only be estimated with $HR_\tau$ if the potential outcomes are independent ($T^0 \perp\!\!\!\perp T^1$). Martinussen et al. (2020) also defines a conditional hazard ratio which is equivalent to the *causal* conditional hazard ratio if the potential outcomes are independent conditional on measured covariates ($T^0 \perp\!\!\!\perp T^1 | X$):

$$
HR(\tau|x) = \frac{h^1(t|x)}{h^0(t|x)} = \frac{P(T^1 = \tau | T^0 \geq \tau, T^1 \geq \tau, C \geq \tau, X = x)}{P(T^0 = \tau | T^0 \geq \tau, T^1 \geq \tau, C \geq \tau, X = x)} = \frac{P(T^1 = \tau | T^1 \geq \tau, C \geq \tau, X = x)}{P(T^0 = \tau | T^0 \geq \tau, C \geq \tau, X = x)}
\tag{8}
$$

However, Martinussen et al. (2020) stresses that both assumptions are both unrealistic and untestable. Thus, methods that depend on this assumption should also be examined with a sensitivity analysis (Axelrod and Nevo, 2022).

## A.8 Other effect measures

Researchers have recommend the use of treatment effect measures that are not conditioned on survival (Hernán, 2010; Martinussen, 2021):

- *Difference in survival times*: $S^1(\tau|x) - S^0(\tau|x)$

- *Difference in restricted mean survival time (RMST)*: $\sum_{\tau_k \leq \tau^*} (S^1(\tau_k|x) - S^0(\tau_k|x))(\tau_k - \tau_{k-1})$
  The RMST is the expected time-to-event conditioned on a specified time horizon, $\tau^*$. For example, if time-to-event outcome is death, $RMST(\tau^*)$ can be interpreted as the $\tau^*-$year life expectancy.

- *Relative risk function (Martinussen et al., 2020)*: $RR(\tau) = \frac{P(T^1 \geq \tau)}{P(T^0 \geq \tau)}$

### A.9 MITIGATING CONFOUNDING AND SELECTION BIAS

Confounding and selection bias can lead to (1) a lack of exchangeability, complicating causal effect estimation, and (2) covariate shift, which can lead to bias in model estimation, particularly if the model is mis-specified (Shimodaira, 2000). If *conditional exchangeability* is satisfied, conditional survival models are causal, and heterogeneous treatment effects calculated from causal conditional survival models can be considered valid (Hernan and Robins, 2023). To recover average treatment effects under conditional exchangeability, various methods such as stratification, standardization, and inverse propensity weighting can be used (Hernan and Robins, 2023). However, it remains difficult to adjust for potential issues of covariate shift, particulary in the face of hetergeneous, time-varying treatment effects. Novel methods have been proposed that rely on the learning of balanced representations to overcome these issues (Chapfuwa et al., 2021; Curth et al., 2021a), but it remains an open area for further study.

## B ADDITIONAL EXPERIMENTAL DETAILS

### B.1 SYNTHETIC DATA GENERATION ALGORITHM

---

**Algorithm 1:** Generating synthetic or semi-synthetic data

---

**Input:** Covariate features $\{x_i\}_{i=1}^N$, generated synthetically or adopted from a real-world dataset, treatment assignment mechanism $\alpha(x)$, hazard functions $h^a(\tau|x)$, censoring hazard functions $h_c^a(\tau|x)$ for treatments $a \in \{0, 1\}$, maximum duration $T_{max}$

**Output:** Semi-synthetic dataset $\mathcal{D} = \{x_i, a_i, y_i, \delta_i\}_{i=1}^N$

$\mathcal{D} \leftarrow \varnothing$ **for** $i \in [N]$ **do**

    $a_i \sim Ber(\alpha(x_i))$;

    $t_i \sim h^{a_i}(\tau|x_i)$ ;      /* sample event time using inverse transform sampling */

    $c_i \sim h_c^{a_i}(\tau|x_i)$ ;      /* sample censoring time */

    **if** $t_i \leq c_i$ **then**

        $y_i \leftarrow t_i$;

        $\delta_i \leftarrow 1$;

    **end**

    **else**

        $y_i \leftarrow c_i$;

        $\delta_i \leftarrow 0$;

    **end**

    **if** $y_i > T_{max}$ **then**

        $y_i \leftarrow T_{max}$;

        $\delta_i \leftarrow 0$;

    **end**

    $\mathcal{D} \leftarrow \mathcal{D} \cup \{x_i, a_i, y_i, \delta_i\}$

**end**

---

### B.2 SYNTHETIC COVARIATE GENERATION

We sample the synthetic covariates from a multivariate normal distribution with mean vector $\mathbf{0}$ and covariance matrix $\Sigma$, where all variables are correlated by the same value $\rho$. The distribution is given by $X \sim \mathcal{N}(\mathbf{0}, \Sigma)$, where the covariance matrix $\Sigma$ is defined as $\Sigma_{ij} = 1$ if $i = j$, $\rho_c$ if $i \neq j$, representing a covariance matrix with diagonal elements 1 and off-diagonal elements $\rho_c$. In all

experiments, the synthetic datasets consists of 10000 samples and 10 covariates, normalized to a $(0, 1)$ scale.

## B.3 REAL-WORLD DATASETS

| Name | # Samples | # Covariates | Treatment | Outcome |
|---|---|---|---|---|
| Twins (Almond et al., 2004) | 10536 | 44 | Heavier birth weight | Survival |
| TCGA (Weinstein et al., 2013) | 9695 | 100 | Radiation therapy | Survival |
| IHDP (Hill, 2011) | 985 | 26 | Treatment (RCT) | IQ score (36 months) |
| News (Johansson et al., 2016) | 10000 | 50 | N/A | N/A |
| SUPPORT (Connors et al., 1995) | 9105 | 27 | N/A | Survival |
| METABRIC (Curtis et al., 2012) | 1980 | 25 | Chemotherapy | Survival |

Table 2: Real-world datasets

| Scn. | Survival model | Description |
|---|---|---|
| 2 | $h^a(t\|x) = 0.5 \exp(-2 + a + x_0)$ | Constant-time, heterogeneous hazards |
| 3 | $h^a(t\|x) = 0.5 \exp(-2 + a \cdot x_0 + x_0)$ | Constant-time, heterogeneous HRs |
| 1 | $h^a(t\|x) = 0.3 \exp(0.1a + 0.3a \cdot x_0 + 0.2a \cdot x_1)$ | Well-specified to Cox PH |
| | $h_c^a(t\|x) = 0.2 \exp(0.1x_2)$ | |
| 2 | $h^a(t\|x) = \begin{cases} 0.3 \exp(0.1a + 0.1a \cdot x_0), & \text{if } x_0 > 0 \\ 0.3 \exp(0.1a + 0.5a \cdot x_0), & \text{otherwise} \end{cases}$ | Mis-specified to Cox PH |
| 3 | $h^a(t\|x) = 0.5 \exp(0.1a + 0.5a \cdot t)$ | Well-specified to Cox TV |
| | $h_c^a(t\|x) = 0.3 \exp(0.01x_2^2 \cdot t)$ | |
| 4 | $h^a(t\|x) = \begin{cases} 0.5 \exp(0.1a + 0.05a \cdot t), & \text{if } t > 10 \\ 0.5 \exp(0.1a + 0.01a \cdot t), & \text{otherwise} \end{cases}$ | Mis-specified to Cox TV |
| 5 | $h^a(t\|x) = 0.8 \exp(0.8a - 0.05a \cdot t)$ | Well-specified to Cox TV, decreasing HR |
| 6 | $\begin{cases} h^a(t\|x) = 0.8 \exp(0.8a - 0.05a \cdot t), & \text{if } t > 10 \\ h^a(t\|x) = 0.8 \exp(0.8a - 0.01a \cdot t), & \text{otherwise} \end{cases}$ | Mis-specified to Cox TV, decreasing HR |
| 7 | $h^a(t\|x) = 0.5 \exp(0.3a \cdot x_0 - 0.1a \cdot t + 0.1x_2 \cdot t)$ | TV and heterogeneous, increasing HR |
| 8 | $h^a(t\|x) = 0.5 \exp(-0.3a \cdot x_0 + 0.1a \cdot t + 0.01x_2 \cdot t)$ | TV and heterogeneous, decreasing HR |
| 9 | $h^a(t\|x) = 0.3 \exp(0.2a \cdot x_0 \cdot \log(t + 0.01) + 0.2a)$ | TV heterogeneously |
| 10 | $h^a(t\|x) = 0.3 \exp(0.2a \cdot x_0 \cdot \log(t + 0.01) - 0.1a \cdot x_1 \cdot \log(t + 0.01) + 0.2a)$ | TV heterogeneously |
| | $h_c^a(t\|x) = 0.1 \exp(0.2x_0)$ | |

Table 3: Experiments: synthetic data generating functions

## B.4 METHOD DETAILS

Table 4 summarizes the attributes of the different models evaluated in the paper.

| Name | Time Scale | Description | Implementation |
|---|---|---|---|
| Logistic Hazard | Discrete | Modified logistic regression | scikit-learn |
| Cox PH | Continuous | Semi-parametric | lifelines |
| Extended Cox | Continuous | Semi-parametric | lifelines |
| Random Survival Forests | Continuous | Modified random forest | Chemotherapy |
| DeepSurv | Continuous | Neural network | PyCox |
| Cox-Time | Continuous | Neural network | PyCox |
| DeepHit | Discrete | Neural network | PyCox |

Table 4: Methods

| Mis-specification | LH | CPH | CTV | RSF | DS | CT | DH |
|---|---|---|---|---|---|---|---|
| Heterogeneous HRs (covariate-treatment interaction) | ✖ | ✖ | ✖ | | | | |
| Time-varying homogeneously (treatment-time interaction) | ✖ | ✖ | | | ✖ | | |
| Time-varying heterogeneously (covariate-treatment-time interaction) | ✖ | ✖ | ✖ | | ✖ | | |
| Non-linearity over covariates | ✖ | ✖ | ✖ | | | | |
| Non-linearity over time | ✖ | ✖ | ✖ | | | | |
| Covariate-covariate interactions | ✖ | ✖ | ✖ | | | | |
| Unmeasured variables | ✖ | ✖ | ✖ | ✖ | ✖ | ✖ | ✖ |

Table 5: Types of mis-specification. ✖'s mark where a type of mis-specification applies to a method.

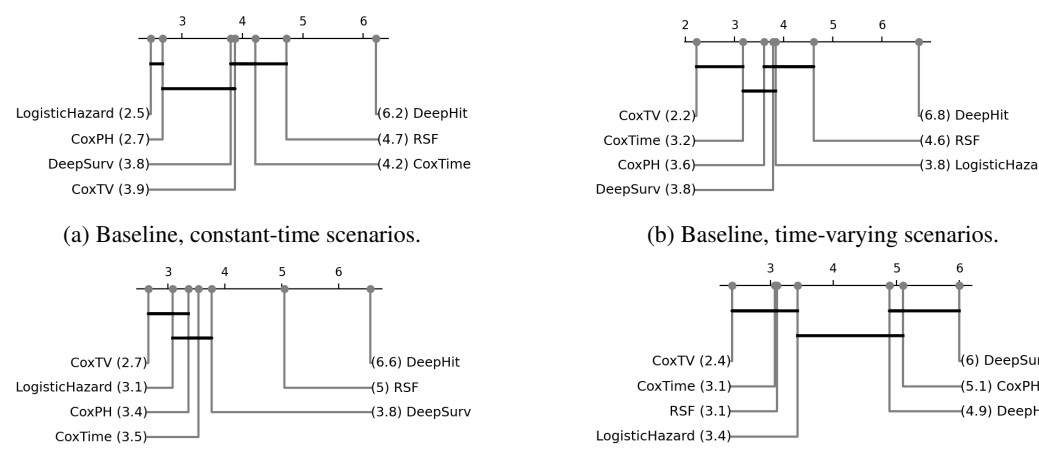

(a) Baseline, constant-time scenarios.

(b) Baseline, time-varying scenarios.

(c) Mis-specified, constant-time scenarios.

(d) Mis-specified, time-varying scenarios.

Figure 8: Critical difference diagrams of average ranks (based on MALE averaged over the 76th to 99th percentile of survival times).

## C  ADDITIONAL EXPERIMENTAL RESULTS

### C.1  SUMMARY OF RESULTS

### C.2  ADDITIONAL HEATMAPS FOR BASELINE AND MIS-SPECIFIED SCENARIOS.

Figure 10 contains additional MALE heatmaps for several baseline and misspecified scenarios.

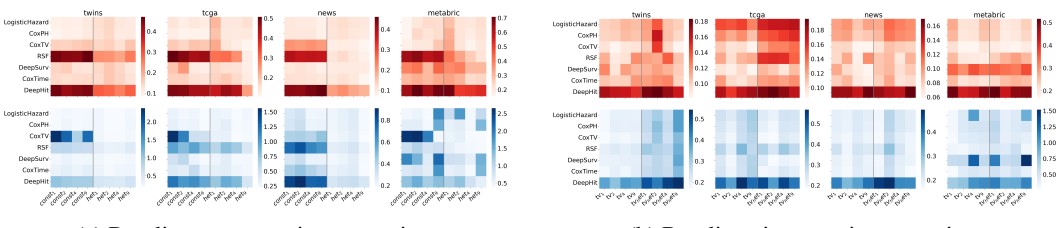

(a) Baseline, constant-time scenarios.

(b) Baseline, time-varying scenarios.

Figure 9: RMSE heatmaps comparing model performance (heatmap rows) over baseline scenarios (heatmap columns) on different datasets (columns). For each dataset, average RMSE is reported up to the 75th percentile of survival times (top row) and 76th to 99th percentile of survival times (bottom row). Grey lines group variations of similar scenarios.

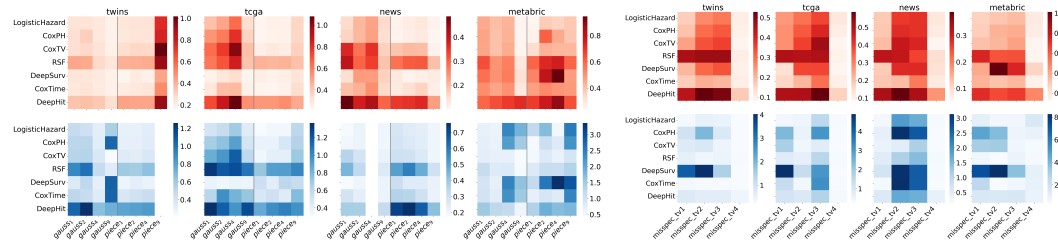

(a) Mis-specified, constant-time scenarios.     (b) Mis-specified, time-varying scenarios.

Figure 10: RMSE heatmaps comparing model performance (heatmap rows) over baseline scenarios (heatmap columns) on different datasets (columns). For each dataset, average RMSE is reported up to the 75th percentile of survival times (top row) and 76th to 99th percentile of survival times (bottom row). Grey lines group variations of similar scenarios. Note that heatmap scales are all different.

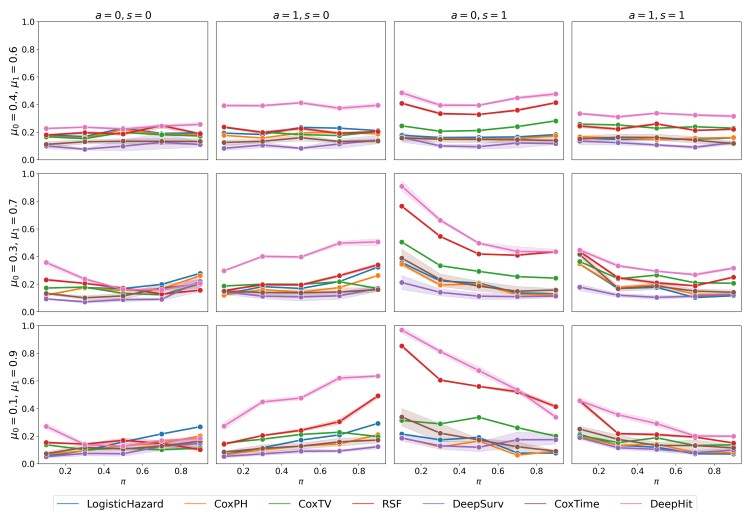

Figure 11: Fairness experiment with measured covariates.

## C.3 SUBGROUP FAIRNESS: SURVIVORSHIP BIAS

Figures 11 and 12 give the results for the subgroup fairness experiments.

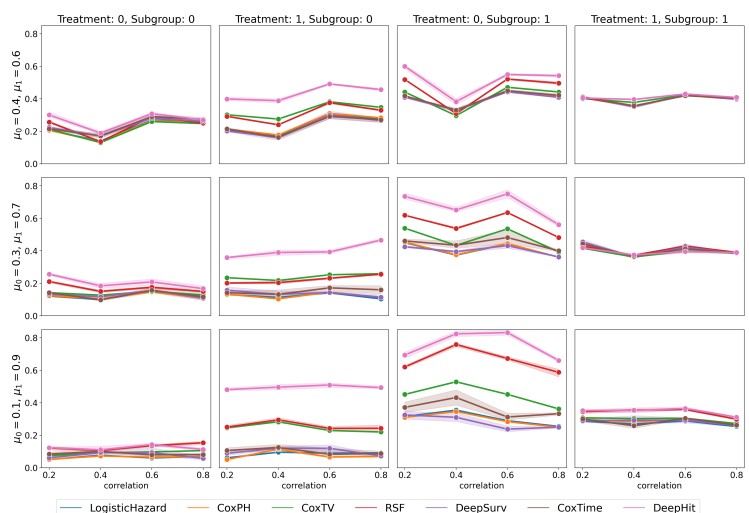

Figure 12: Fairness experiment with unmeasured covariates.

# D    EVALUATION METRICS

## D.1    MEAN ABSOLUTE LOGIT ERROR (MALE)

In the following subsection, for notational clarity we drop the conditioning on the treatment $a$ and covariates $x$.

**Theorem D.1.** MALE *is a strictly proper scoring rule, i.e., it is minimized if and only if* $h(\tau) = \hat{h}(\tau)$ *for all* $\tau$.

*Proof.* It is clear that MALE is minimized iff it is minimized termwise in $\tau$. In this case, for all $\tau$, we have

$$\left| \log \frac{h(\tau)}{1 - h(\tau)} - \log \frac{\hat{h}(\tau)}{1 - h(\tau)} \right| = 0 \iff \frac{h(\tau)}{1 - h(\tau)} = \frac{\hat{h}(\tau)}{1 - h(\tau)}$$

$$\iff h(\tau) - h(\tau)\hat{h}(\tau) = \hat{h}(\tau) - h(\tau)\hat{h}(\tau)$$

$$\iff h(\tau) = \hat{h}(\tau),$$

i.e., the estimated hazard is equal to the ground truth as desired. □

**Theorem D.2.** *Let the ordered discrete time intervals be* $\tau_1, \ldots, \tau_k$. *Define* $\mathbb{P}(\tau_k) = \prod_{i=1}^{k-1}(1 - h(\tau_i))h(\tau_k)$ *be the probability that a unit with treatment* $a$ *and features* $x$ *fails in* $\tau_i$, *according to the ground truth hazard function* $h$. *Define* $\hat{\mathbb{P}}(\tau_k)$ *analogously for the estimated hazard* $\hat{h}$. *Then* $\left| \log \frac{\hat{\mathbb{P}}(\tau_k)}{\mathbb{P}(\tau_k)} \right| \leq \sum_{i=1}^{k} \mathrm{ALE}(\tau_k)$ *for all* $k$.

*Proof.* First, we observe the following inequality: for any $p, q \in (0, 1)$, we have

$$\left| \log \frac{p}{q} \right|, \quad \left| \log \frac{1 - q}{1 - p} \right| \leq \left| \log \frac{p}{q} + \log \frac{1 - q}{1 - p} \right|. \tag{9}$$

To see this, assume first that $p \geq q$. Then $1 - q \geq 1 - p$, so all of the individual log terms are positive and the inequality is trivial. When $p < q$, all of the individual log terms are negative and the same inequality holds in terms of the absolute values.

With this inequality in hand, a direct computation shows that

$$\left| \log \frac{\hat{\mathbb{P}}(\tau_k)}{\mathbb{P}(\tau_k)} \right| = \left| \log \frac{\prod_{j=1}^{i-1}(1-\hat{h}(\tau_i))\hat{h}(\tau_k)}{\prod_{j=1}^{i-1}(1-h(\tau_i))h(\tau_k)} \right|$$

$$\leq \sum_{j=1}^{i-1} \left| \log \frac{1-\hat{h}(\tau_i)}{1-h(\tau_i)} \right| + \left| \log \frac{\hat{h}(\tau_k)}{h(\tau_k)} \right|$$

$$\leq \sum_{j=1}^{i} \left| \log \frac{1-\hat{h}(\tau_i)}{1-h(\tau_i)} + \log \frac{h(\tau_i)}{\hat{h}(\tau_i)} \right|.$$

The final inequality holds by applying inequality equation 9 with $p = \hat{h}(\tau_i)$ and $q = h(\tau_i)$ for each $j = 1, \ldots, i$. This final bound is precisely $\sum_{i=1}^{k} \mathrm{ALE}(\tau_i)$, as desired. □

**Theorem D.3.** *Let* $\mathrm{MSE} = \alpha$. *For any* $\alpha > 0$, *we have* $\sup \left| \log \frac{\hat{\mathbb{P}}(\tau)}{\mathbb{P}(\tau)} \right| = \infty$, *where the supremum is taken over all* $h, \hat{h}$ *such that the MSE of* $\hat{h}$ *with respect to* $h$ *is at most* $\alpha$. *In other words, the survival probabilities cannot be bounded in terms of the MSE.*

*Proof.* We give two simple constructions, one in which one in which the hazards are allowed to be equal to 0 and one in which all hazards must be contained in $(0, 1)$.

For the first case, if we define $h(\tau_1) = 0$ and $\hat{h}(\tau_1) = \sqrt{\alpha}$ then the MSE is equal to $\alpha$ but $|\log(\hat{\mathbb{P}}(\tau_1)/\mathbb{P}(\tau_1))| = \infty$.

For the second case, define $h(\tau_1) = \varepsilon\sqrt{\alpha}$ and $\hat{h}(\tau_1) = (1+\varepsilon)\sqrt{\alpha}$, where $\varepsilon > 0$ is assumed to be very small (so that $h(\tau_1), \hat{h}(\tau_1) < 1$). Observe that the MSE is equal to $\alpha$, but $|\log(\hat{P}(\tau_1)/\mathbb{P}(\tau_1))| = \frac{1+\varepsilon}{\varepsilon}$. Taking $\varepsilon \to 0$ gives the desired result. □

### D.2 OTHER EVALUATION METRICS

**Brier score** The Brier score is a time-dependent measure of the quality of an estimated survival function, which computes the squared error of the survival probability predicted by the model vs. the binary label of whether or not the datapoint being evaluated has failed by the specified time. This squared loss is reweighted to account for censoring, and a time-independent version of the Brier score (called the integrated Brier score or IBS) is given by averaging the time-dependent score over the desired time interval. While the IBS is a proper scoring rule in the absence of censoring, Rindt et al. (2022) showed that it may *not* be a proper scoring rule when the censoring mechanism depends on the covariates. We refer the reader to Section 3.1 of Rindt et al. (2022) for more details.

**1-Calibration** 1-calibration compares predicted survival probabilities with outcomes in the data and measures how well the two agree. Specifically, the data are binned into pre-specified bins based on their predicted survival probabilities at the time they experienced an event. The actual number of failures is compared to the expected number of failures (according to the model's predicted failure probabilities) for each bin. Under the null hypothesis that the model's probabilities are correct, these deviations can be used to define a test statistic which follows a $\chi^2$ distribution, which can be used to construct a hypothesis test for the calibration of the model. We refer the reader to Section 3.3 of Haider et al. (2018) for complete details. As mentioned by Haider et al. (2018) in the following section of their paper (Section 3.4), 1-calibration is not effective for ranking multiple models beyond suggesting some of the models are calibrated (high p-value) and others are not (low p-value).

**D-Calibration** D-calibration is similar to 1-calibration in that it compares estimated failure probabilities to the expected number of events that would occur if these probabilities were accurate. However, instead of just comparing the predicted probabilities at the event time for each datapoint, D-calibration seeks to measure the goodness of fit of the *entire survival distribution* predicted by the

model. This is again accomplished using a probability binning procedure followed by a hypothesis test. We refer the reader to Section 3.5 of Haider et al. (2018) for the complete details of this metric. While D-calibration does give a more nuanced evaluation of the calibration of a survival model as compared to 1-calibration, it suffers from the same problem, namely, it cannot be used to rank many survival models beyond suggesting that some are poorly calibrated (low p-value) while others are not (high p-value).

