# OpenReview forum: "Benchmarking Survival Models: Treatment Effects, Bias, and Equity"
_ICLR.cc/2025/Conference — ICLR 2025 Conference Withdrawn Submission_

### Official Review · Reviewer_YCdF · 2024-10-21

**Soundness:** 3
**Presentation:** 1
**Contribution:** 1
**Rating:** 1
**Confidence:** 4

**Summary:**

This paper empirically examined the performance of several estimators focusing on time-to-event outcomes in causal settings. In detail,
1. The estimators examined include the parametric logistic hazard model, the semi-parametric Cox proportional hazards model (CoxPH), the time-varying Cox model, Random survival forests, and neural network methods DeepSurv, CoxTime, and DeepHit;
2. The estimators were evaluated on several real datasets, varying in number of covariates and sample sizes. The treatment, outcome, and censoring time, given covariates, were however synthetically generated;
3. The estimand is discrete time hazard function and the evaluation metric is the "mean absolute logit error (MALE)", ranked by critical difference;
4. Different sources of bias are considered: model bias (time-varying treatment effect, non-linearity), confounding bias (confounding strength, unmeasured confounders), survivorship bias linked to protected attributes.

**Strengths:**

The paper touched on various practical and commonly used estimators in survival analysis with a rich consideration on sources of biases.

**Weaknesses:**

1. The paper is not well motivated. Why should readers be interested and what can they learn? Theoretically, we all know that certain estimators are subject to bias if either model is wrong or identifying assumption is violated, so I believe the additional value of this paper shall be from empirical level? Then can we say some estimators are robust to some kind of sources of bias from paper? Why should we care the biases from this paper instead of using sensitivity analyses provided in the literature? I think this also results from the 2nd point below that I am not sure what the conclusion is.

2. The presentation needs greatly improving. After reading through the paper, I don't get what the take-away and I don't think the paper is well structured. The current draft has only observations of estimators performance scattered everywhere. Authors can spend efforts on what the conclusion the paper is. Provide a unified and principal conclusion to empirical strengths and weaknesses of the estimators considered. This can be a table comparing the strengths and weaknesses of each estimator across different scenarios, or a dedicated "Conclusions" section that synthesizes the main takeaways.

3. Other minor issues include
 a. Figure 1 three pictures are the same;
 b. Page 4, line 193, "Due to space constraints, the relevant figure (Fig. 2) can be found in App. A.3." The fig.2 is in fact on the same page;
 c. Page 6, line 301, "The characteristics of the real-world datasets are summarized in Table 2, with more details in App B.3". In fact, there is nothing more than Table 2  in App B.3. Same page, line 304, "confonding", misspelling;
 d. Page 8, line 402, "Fig ??"
These implied that this paper has not been scanned carefully by the authors. Please conduct a thorough proofreading pass, paying particular attention to figure references and consistency between the main text and appendices.

**Questions:**

Things where a response from the author can change my opinion: Be specific to motivate the readers on what they can learn from your paper and why they should care the results. Give more motivations and conclusions like I mentioned in the weakness might help on this.

Confusion Clarifying:
Is the estimand h(\tau|a, x) changing when you changes the indices of the covariates that affects the hazard function in your simulation?

Address a limitation:
Though using hazard function as the estimand, the paper mentioned multiple times that it is controversial because of "the built-in
selection bias of hazard ratios". And the papers used it because it is commonly reported in practice. I in fact found the following "On defense of the hazard ratio" A Ying, R Xu arXiv preprint arXiv:2307.11971 to be along the line of that discussion but supporting and defending the usage of hazard function, which shall be helpful to strenghen the paper.

---

### Official Review · Reviewer_4ze7 · 2024-10-22

**Soundness:** 1
**Presentation:** 1
**Contribution:** 1
**Rating:** 3
**Confidence:** 4

**Summary:**

This paper provides an extensive examination of the challenges and unrealistic assumptions inherent in causal inference within time-to-event problem settings. The authors propose a benchmarking framework for survival models, introducing a novel evaluation metric. Based on empirical performance, they offer insights and recommendations for benchmarking survival analysis models across various scenarios, especially when certain assumptions or conditions may be violated by the models.

**Strengths:**

The paper provides a comprehensive literature review on the key biases and challenges that arise in causal inference with time-to-event data settings. It studies a critical problem -- the lack of unbiased and effective evaluation metrics in causal survival analysis settings -- which has not received sufficient attention and remains unsolved.

**Weaknesses:**

While the paper is set to identify the challenges in causal survival analysis, it is constrained by a lack of technical novelty and several other issues. The only original contribution -- the proposed MALE metric -- is limited to synthetic datasets, restricting its practical applicability. Although the paper aims to serve as a benchmarking study, it includes only seven baseline models, predominantly Cox-based, with the most recent from 2019. Additionally, the manuscript appears underprepared, exhibiting numerous typos, missing details, and misused figures.

**Lack of Technical Novelty and Issues with the Proposed Metric:**
1. **Insufficient Theoretical Insight for the MALE Metric:** The proposed MALE metric is introduced briefly (lines 265-285), using less than half a page, without providing substantial theoretical justification or demonstrating how it effectively addresses the discussed challenges in causal survival analysis.
2. **Limited Practical Value of the Evaluation Framework:** The evaluation framework relies on the ground truth hazard function to calculate the MALE score, which is only available in oversimplified synthetic datasets. This dependency significantly limits the framework's applicability to real-world scenarios where the ground truth is unknown.
3. **Weak Motivation and Comparison with Existing Metrics:**: The authors assert that existing benchmarks do not estimate the fidelity to the ground truth hazard or survival models (lines 74-76). However, if the goal is to simply estimate the error between the true hazard (resp, survival) function and the predicted hazard (resp, survival) function in the synthetic dataset (despite its useless in the real-world scenario, as discussed above), many papers have such evaluation metrics (e.g., Table 1 & 2 in [1], Survival-$l_1$ in [2], RMSE of predicted and true survival function in [3], etc.). The paper does not sufficiently compare the proposed metric with any of those or argue how the proposed MALE metric offers advantages over these existing measures.

**Insufficient Benchmarking Comparison**
1. **Outdated and Limited Baseline Models:** The study includes only seven baseline models, mainly focusing on Cox-based methods, with the latest model (CoxTime) introduced in 2019. Given the numerous state-of-the-art survival models developed in recent years, incorporating more advanced methods would enhance the comprehensiveness of the benchmarking.
2. **Absence of Causal Survival Models in Comparison:** Despite aiming to compare traditional and modern survival models in causal settings, the paper does not benchmark any causal survival models, such as CSA [4], SurvITE [3], CMHE [5], DNMC [6], or compCATE [7].
3. **Oversimplification in Synthetic Data Generation:** The synthetic datasets used are overly simplistic, with the most complex modeling involving a cubic transformation of a single feature or a linear combination of at most two features. This simplicity may not capture the complexity of real-world data and limits the generalizability of the findings.


**Factual Mistakes and Errors**:
1. **Incorrect Definition of the Hazard Function (Line 107):** The paper states that "In discrete-time, the hazard function is the probability that the individual will experience the event outcome in a given interval of time," which is incorrect. The correct definition is that the hazard function in discrete time is the probability that an individual will experience the event in a given interval, given that they have not experienced it in any prior intervals.
2. **Misclassification of the MALE Metric as a Calibration Metric (Lines 268-269):** The authors describe the MALE metric as a calibration metric. However, since it uses the hazard function -- which can exceed 1 in continuous models --it cannot be interpreted as a probability prediction, and therefore, it does not measure calibration.
3. **Incorrect Description of Non-informative Censoring (Line 306):** The data generation procedure is labeled as "non-informative censoring," but in Scenario 10 (Table 3), the hazard functions for both event and censoring times depend on $x_0$ (meaning $T\perp C \mid X$). This violates the definition of non-informative censoring ("Non-informative censoring occurs if the distribution of survival times (T) provides no information about the distribution of censorship times (C), and vice versa." [8], meaning $T\perp C$).
4. **Significant issues with figures**
   - The three subfigures in Figure 1 are identical, making them uninformative. Additionally, the figure is not self-contained; the meanings of the red, dashed, and solid arrows are not explained. The random variable $Z$ is not introduced when the figure is first mentioned (line 151) or in the caption, but only later in line 189.
   - The subtitles in Figure 2 are incorrect. Based on the descriptions in lines 194-205, the subtitles should be: (a) Direct Effect, (b) Indirect Effect via Covariate Shift, and (c) Indirect Effect via Unmeasured Variables.
   - The text refers readers to Figure 2 in Appendix A.3 due to space constraints, but Appendix A.3 on page 16 is empty.
   - Figures 6 and 7 are duplicates, containing identical diagrams and captions.

References:

[1] Tang et al., Survival Analysis via Ordinary Differential Equations, JASA 2022

[2] Foomani et al., Copula-Based Deep Survival Models for Dependent Censoring. UAI 2023

[3] Curth et al., 2021b (cited in the paper)

[4] Chapfuwa et al., 2021 (cited in the paper)

[5] Nagpal et al., Counterfactual Phenotyping with Censored Time-to-Events. KDD 2022

[6] Engelhard and Henao, Disentangling Whether from When in a Neural Mixture Cure Model for Failure Time Data. AISTATS 2022

[7] Curth and van der Schaar, Understanding the Impact of Competing Events on Heterogeneous Treatment Effect Estimation from Time-to-Event Data. AISTATS 2023

[8] Survival analysis: A self-learning text, Kleinbaum and Klein (3rd edition, 2011, Springer), Page 42

**Questions:**

See above.

---

### Official Review · Reviewer_rhri · 2024-11-01

**Soundness:** 3
**Presentation:** 2
**Contribution:** 2
**Rating:** 3
**Confidence:** 4

**Summary:**

In this work, the authors propose a semi-simulation study based on 6 real datasets for survival analysis. The authors frame their evaluation  in a causal and fairness framework, instead of the classical predictive framework. They introduce a calibration metric, the MALE to assess the relative performances of the considered methods, in a wide variety of scenarios (model mis-specification, confounding bias, survivorship bias)

**Strengths:**

This work adresses an important and interesting problem, and proposes a thorough overview of the problem of causal inference with right-censored data, and some of the challenges associated with different types of bias. The simulation study covers several varied datasets. For all topics covered (estimands for causal survival analysis, metrics) the authors provide a thorough overview of the literature.

**Weaknesses:**

There are several limitations to the proposed work
- the objective of the estimation is not clearly state. Here the authors try to estimate the conditional hazard fonction or the conditional survival function. Those are not causal estimands, their difference could be a causal estimand. They mention rightfully that the hazard ratio is not a well-suited target due to problems of selection bias in its definition.
- to estimate a causal estimand, there can be several estimators, like the plug-in g-formula, the inverse-propensity-weighting-score methods, or multiply robust methods. Here it seems that the authors implicitely consider the plug-in g-formula, but do not clearly state it. This in turn hinders a flaw in the evaluation: the final computation of the effect often relies on the estimation of so-called nuisance functions (the survival or hazard functions in this case), that are then aggregated (by taking their difference for example). However, having a better estimation of the nuisance functions does not garantee a better estimation of their difference. This is well documented (Nie, Xinkun, and Stefan Wager. "Quasi-oracle estimation of heterogeneous treatment effects." Biometrika 108.2 (2021): 299-319., Saito, Yuta, and Shota Yasui. "Counterfactual cross-validation: Stable model selection procedure for causal inference models." International Conference on Machine Learning. PMLR, 2020.). The authors need to consider several estimators in addition of several methods to estimate the nuisance parameters (or at least to explicit what they evaluate exactly). Here is an example of work with survival analysis and several estimators Cheng, Chao, et al. "Multiply robust estimation for causal survival analysis with treatment noncompliance." arXiv preprint arXiv:2305.13443 (2023). It is additionally not right to make claims on the ability of the considered methods to measure treatment effects while simply evaluating the calibration of the nuisance functions separately (and not their difference)
- the proposed metric is interesting, however it is limited to cases where the ground truth is known, which is of very limited practical use. The other mentioned metrics, though having limitations, could be interesting, in particular, as one could then compare the behavior of those metrics on real data and actually use this benchmark in practice. The package https://github.com/shi-ang/SurvivalEVAL implements several of those metrics. Additionally, a good calibration does not ensure that you recover the actual distribution, a proper metric is needed for that.
- there is no mention that the code will be made available, it could be interesting that this benchmark can be further re-used by others, otherwise its interest is very limited, and all survival methods cannot be evaluated by you (see https://survival-org.github.io/DL4Survival/)
- several figures are really hard to read
    - figure 1a, b and c are identical. Is it an error? the legend or the text should contain a description of the variables, and a short hint about how each bias occurs in those cases
    - the text says Figure 2 is in the appendices, while it finally made it to the main text
    - Figure 5 (and other similar figures in the appendices) don't have a description of the y axis (the y label describes some setting of the simulation). There is also no mention of the dataset
- broken reference line 402
- missing closing parenthesis line 405

- the conclusion/discussion does not mention all the methods, nor provides clear guidance on which method to use

Overall, the attempt is interesting, but several key pieces are missing to make this work useful in practice.

**Questions:**

- can you be more explicit about the estimand that you consider? and maybe consider several?
- same question for estimators
- can you provide additional metrics? in particular metrics that are applicable without knwoing the ground truth?

---

### Note · Authors · 2024-12-02

**Comment:**

We thank the reviewers for their thorough comments. We agree that our paper could benefit from additional edits beyond the scope of the rebuttal process and are appreciative of the related works brought to our attention. Therefore, we have decided to withdraw the paper from consideration. However, we would like to respond to some of the specific points brought up by the reviewers:

* **Motivation for and usefulness of the framework**: We agree that we can make additional clarifications regarding the express purpose of our framework. To clarify here, we are interested in evaluation of novel methods for survival modelling. In clinical research, a survival model is more clinically informative if the model itself is correct, i.e. calibrated to the true survival distribution. However, it is not possible to know the true survival distributions of purely observational data, so we generate semi-synthetic data to ensure models perform well under a variety of circumstances (e.g. misspecification, confounding, censoring bias). In purely observational data, the presence of unmeasured confounders may affect evaluation. However, we acknowledge the interest in using purely observational data for evaluation and will incorporate this into future iterations of our work.

* **The proposed MALE metric**: We provide theoretical justification for MALE as a metric that is only minimized with the correct true survival distribution in Appendix D.1. We provide theoretical criticism of the use of RMSE in Theorem D.3. While we acknowledge that MALE is only useful with known ground truths, we believe that this is an essential facet of evaluation of survival models.

* **Inclusion of causal survival models**: In this work, we focused on benchmarking a set of widely-used and well-known models to demonstrate the usefulness of the framework. We acknowledge reviewer rhri’s comment that we did not clarify the estimation of our causal estimands. We utilized the S-learner strategy [1], where we train a single model to estimate patient outcomes where treatment is simply another covariate used in the model. The estimated outcomes from this model are then used to calculate treatment effects. However, we agree that this should have been clarified and that it would be beneficial to compare this with other estimation strategies, such as other treatment effect meta-learners [1] or methods that attempt to optimize over estimation of the treatment effect, such as causal survival forests (as suggested by reviewer 4ze7 as well).

[1] Künzel, S. R., Sekhon, J. S., Bickel, P. J., & Yu, B. (2019). Metalearners for estimating heterogeneous treatment effects using machine learning. Proceedings of the national academy of sciences, 116(10), 4156-4165.

**Withdrawal Confirmation:**

I have read and agree with the venue's withdrawal policy on behalf of myself and my co-authors.